# DPSCREEN: Dynamic Personalized Screening

**Kartik Ahuja**
Electrical and Computer Engineering Department
University of California, Los Angeles
ahujak@ucla.edu

**William R. Zame**
Economics Department
University of California, Los Angeles
zame@econ.ucla.edu

**Mihaela van der Schaar**
Engineering Science Department, University of Oxford
Electrical and Computer Engineering Department, University of California, Los Angeles
mihaela.vanderschaar@oxford-man.ox.ac.uk

## Abstract

Screening is important for the diagnosis and treatment of a wide variety of diseases. A good screening policy should be *personalized* to the features of the patient and to the dynamic history of the patient (including the history of screening). The growth of electronic health records data has led to the development of many models to predict the onset and progression of different diseases. However, there has been limited work to address the personalized screening for these different diseases. In this work, we develop the first framework to construct screening policies for a large class of disease models. The disease is modeled as a finite state stochastic process with an absorbing disease state. The patient observes an *external information process* (for instance, self-examinations, discovering comorbidities, etc.) which can trigger the patient to arrive at the clinician earlier than scheduled screenings. The clinician carries out the tests; based on the test results and the external information it schedules the next arrival. Computing the exactly optimal screening policy that balances the delay in the detection against the frequency of screenings is computationally intractable; this paper provides a computationally tractable construction of an approximately optimal policy. As an illustration, we make use of a large breast cancer data set. The constructed policy screens patients more or less often according to their initial risk – it is personalized to the *features* of the patient – and according to the results of previous screens – it is personalized to the *history* of the patient. In comparison with existing clinical policies, the constructed policy leads to large reductions (28-68%) in the number of screens performed while achieving the same expected delays in disease detection.

## 1   Introduction

Screening plays an important role in the diagnosis and treatment of a wide variety of diseases, including cancer, cardiovascular disease, HIV, diabetes and many others by leading to early detection of disease [1]-[3]. For some diseases (e.g., breast cancer, pancreatic cancer), the benefit of early detection is enormous [4] [5]. Because screening – especially screening that requires invasive procedures such as mammograms, CT scans, biopsies, angiograms, etc. – imposes financial and health costs on the patient and resource costs on society, good screening policies should trade off benefit and cost [6]. The *best* screening policies should take into account that the trade-off between benefit and cost should be different for different diseases – but also for different patients – patients whose features suggest that they are at high risk should be screened more often; patients whose features suggest that they are at low risk should be screened less often – and even different for the same individual at different points in time, as the perceived risk for that patient changes. Thus the

best screening policies should account for the disease type and be *personalized* to the features of the patient and to the history of the patient (including the history of screening) [32]. This paper develops the first such personalized screening policies in a very general setting.

A screening policy prescribes what tests should/should not be done and when. Developing personalized screening policies that optimally balance the frequency of testing against the delay in the detection of the disease is extremely difficult for a number of reasons. (1) The onset and progression of different diseases varies significantly across the diseases. For instance, in [7] the development of breast cancer is modeled as a stationary Markov process, in [36] the development of HIV is modeled using a non-stationary survival process and, in [46] the development of colon cancer is modeled as a Semi-Markov process. The test outcomes observed over time may follow a non-stationary stochastic process that depends on the disease process upto that time and the features of the patient [35][36]. Existing works on screening [7] [9] are restricted to Markov disease processes and stationary Markov test outcome models, while this is not the case for many diseases and their test outcomes [10][35]-[37]. (2) The cost of not screening is the delay in detection of disease, which is not known. Hence the decision maker must act on the basis of beliefs about *future* disease states in addition to beliefs about the current disease state. (3) Patients can arrive at the scheduled time but may also arrive earlier on the basis of external information so the decision maker's beliefs must take this external information into account. For instance, external information can be the development of lumps on breasts [25][26], or the development of a comorbidity [33][41]. (4) Given models of the progression of the disease and of the external information, *solving* for that policy is computationally intractable in general.

This paper addresses all of these problems. We provide a computationally effective procedure that solves for an *approximately* optimal policy and we provide bounds for the approximation error (loss in performance) that arises from using the approximately optimal policy rather than the exactly optimal policy. Our procedure is applicable to many disease models such as dynamic survival models [11]-[13][36]-[37], first hitting time models [7][9][14]-[17].

Evaluating a proposed personalized screening policy using observational data is challenging. Observational data does not contain the counterfactuals: we cannot know what would have happened if a patient had been screened more often or an additional test had been performed. Instead, we follow an alternative route that has become standard in the literature [7]-[10]: we learn the disease progression model from the observational data and then evaluate the screening policy on the basis of the learned model. We also account for the fact that the disease model may be incorrectly estimated. We show that if the estimation error and the approximation error are small, then the policy we construct is very close to the policy for the *correctly* estimated model.

In this work, use of a large breast cancer data set to illustrate the proposed personalized screening policy. We show that high risk patients are screened more often than low risk patients (personalization to the features of the patient) and that patients with bad test results are screened more often than patients with good test results (personalization to the dynamic history of the patient). The effect of these personalizations is that, in comparison with existing clinical policies, the policy we construct leads to large reductions (28-68%) in screening while achieving the same expected delays in disease detection. To illustrate the impact of the disease on the policy, we carry out a synthetic exercise across diseases, one for which the delay cost is linear and one for which the delay cost is quadratic. We show that the regime of operation (frequency of tests vs expected delay in detection) for the policies for the two costs are significantly different, thus highlighting the importance of choice of costs.

## 2   Model and Problem Formulation

**Time** Time is discrete and the time horizon is finite; we write $\mathcal{T} = \{1, ..., T\}$ for the set of time slots.

**Patient Features** Patients are distinguished by a (fixed) *feature* $x$. We assume that the features of a patient (age, sex, family history, etc.) are observable and that the set $X$ of all patient features is finite.

**Disease Model** We model the disease in terms of the *(true physiological) state*, where the state space is $\mathcal{S}$. The disease follows a finite state stochastic process; $\mathcal{S}^T$ is the space of *state trajectories*. The probability distribution over trajectories depends on the patient's features; for $\vec{s} \in \mathcal{S}^T$, $x \in X$ we write $Pr(\vec{s}|x)$ for the probability that the state trajectory is $\vec{s}$ given that the patient's features are $x$. We distinguish one state $D \in \mathcal{S}$ as the *disease state*; the disease state $D$ is absorbing.[1] Hence

$Pr(s(t) = D, s(t') \neq D) = 0$ for every time $t$ and every time $t' > t$. The true state is hidden/not observed.[2]

Our stochastic process model of disease encompasses many of the disease models in the literature, including discrete time survival models. The (discrete time) Cox Proportional Odds model [11], for instance, is the particular case of our model in which there are two states (Healthy $H$ and Disease $D$) and the probability distribution over state trajectories is determined from the hazard rates. To be precise: if $\vec{s}$ is the state trajectory for which the disease state first occurs at time $t_0$, so that $s(t) = H$ for $t < t_0$ and $s(t) = D$ for $t \geq t_0$, $\lambda(t|x)$ is the hazard at time $t$ conditional on $x$, then $Pr(\vec{s}|x) = [1 - \lambda(1|x)] \cdots [1 - \lambda(t_0 - 1|x)][\lambda(t_0|x)]$ and $Pr(\vec{s}|x) = 0$ for all trajectories not having this form. Similar constructions show that other dynamic survival models [14]-[17] [10][37] also fit in the rubric of the general model presented here.[3]

**External Information**  The clinician performs tests that are informative about the patient's true state; in addition, external information may also arrive (for instance, patient self-examines breasts for lumps, patient discovers comorbidities, etc.). The patient observes an external information process modeled by a finite state stochastic process with state space $\mathcal{Y}$; the information at time $t$ is $Y(t) \in \mathcal{Y}$ (for instance, $\mathcal{Y} = \{\text{Lump}, \text{No Lump}\}$). If the patient visits clinician at time $t$, then this external information $Y(t)$ arrives to the clinician. $Y(t)$ may be correlated with the patient's state trajectory through time $t$ and the patient's features; we write $Pr(Y(t) = y|\vec{s}(t), x)$ for the probability that the external information at time $t$ is $y \in \mathcal{Y}$, conditional on the state trajectory through time $t$ and features $x$. We assume that at each time $t$ the external information $Y(t)$ is independent of the past observations conditional on the state trajectory through time $t$, $\vec{s}(t)$, and features $x$.

**Arrival**  The patient visits the clinician at time $t$ if either (a) the information process $Y(t)$ exceeds some threshold $\tilde{y}$ or (b) $t$ is the time for the next recommended screening (determined in the Screening Policies described below). The first visit of the patient to the hospital depends on the screening policy and the patient's features (See the description below). If the patient visits the clinician at time $t$, the clinician performs a sequence of tests and observes the results. For simplicity of exposition, we assume that the clinician performs only a single test, with a finite set $\mathcal{Z}$ of outcomes. We write $Pr(Z(t) = z|\vec{s}(t), x)$ as the probability that test performed at time $t$ yields the result $z$, conditional on the (unobserved) state trajectory and the patient's features. We assume that the current test result is independent of past test results, conditional on the state trajectory and patient features. We also assume that current test result is independent of the external information conditional on the state trajectory through time $t$ and the patient features. These assumptions are standard [7] [36]. We adopt the convention that $z(t) = \emptyset$ if the patient does not visit the clinician at time $t$ so that no test is performed. If the test outcome $z \in \mathcal{Z}^+ \subset \mathcal{Z}$, then the patient is diagnosed to have the disease. We assume that there are no false positives. If a patient is diagnosed to be in the disease state, then screening ends and treatment begins.

**Screening Policies**  The *history* of a patient through time $t$ consists of the trajectories of external information, test results and screening recommendations through time $t$. Write $\mathcal{H}(t)$ for the set of histories through time $t$ and $\mathcal{H} = \bigcup_{t=0}^{T} \mathcal{H}(t)$ for the set of all histories. By convention $\mathcal{H}(0)$ consists only of the empty history. A *screening policy* is a map $\pi : X \times \mathcal{H} \to \{1, \ldots, T\} \cup \{D\}$ that specifies, for each feature $x$ and history $h$ either the next screening time $t^+$ or the decision that the patient is in the disease state $D$ and so treatment should begin. A screening policy $\pi$ begins at time $0$, when the history is empty, so $\pi(x, \emptyset)$ specifies the first screening time for a patient with features $x$. (For riskier patients, screening should begin earlier.) Write $\Pi$ for the space of all screening policies.

**Screening Cost**  We normalize so that the *cost of each screening* is 1. (We can easily generalize to the more general setting in which the clinician decides from multiple tests [50], and different tests have different costs.) The cost of screening is a proxy for some combination of the monetary cost, the resource cost and the health cost to the patient. We discount screening costs over time so if $\mathcal{T}_s$ is the set of times at which the patient is screened then the *screening cost* is $\sum_{t \in \mathcal{T}_s} \delta^t$, where $\delta \in (0, 1)$.

**Delay Cost** If disease first occurs at time $t_D$ (the *incidence time*) but is detected only at time $t_d > t_D$ (the *detection time*) then the patient incurs a delay cost $C(t_d - t_D; t_D)$. If the disease is never detected the delay cost is $C(T - t_D; t_D)$. We assume that the delay cost function $C : \{1, \ldots, T\} \times \{1, \ldots, T-1\} \to (0, \infty)$ is increasing in the first argument (the *lag* in detection) and decreasing in the second argument (the incidence time). The cost of delay is 0 if disease never occurs or occurs only at time $t = T$. Note that as soon as the disease is detected screening ends and treatment begins; in particular, there is a single unique time of incidence and a single unique time of detection. We allow for general delay costs because the impact of early/late detection on the probability of survival/successful treatment is different for different diseases.

**Expected Costs** If the patient features are $x \in X$ then every screening policy $\pi \in \Pi$ induces a probability distribution $Pr(\cdot|x, \pi)$ on the space $\mathcal{H}(T)$ of all histories through time $T$ and in particular induces probability distributions $\sigma = Pr(\cdot|x, \pi)$ on the families $\mathcal{T}_s \subset 2^{\{1, \ldots, T-1\}}$ of screening times and $\beta = Pr((\cdot, \cdot)|x, \pi)$ on the pairs $(t_D, t_d)$ of incidence time and detection time. The *expected screening cost* is $E_\sigma\left[\sum_{t \in \mathcal{T}_s} \delta^t\right]$ and the *expected delay cost* is $E_\beta\left[C(t_d - t_D, t_D)\right]$. We provide a graphical model for the entire setup in the Appendix B of the Supplementary Materials.

**Optimal Screening Policy** The objective of the screening policy is to minimize a weighted sum of the screening cost and the delay cost; i.e. the optimal screening policy is defined by

$$\arg\min_{\pi \in \Pi}\left\{(1 - w)\, E_\sigma\left[\sum_{t \in \mathcal{T}_s} \delta^t\right] + w E_\beta[C(t_d - t_D, t_D)]\right\} \tag{1}$$

The weight $w$ reflects social/medical policy; for instance, $w$ might be chosen to minimize cost subject to some accepted tolerance in delay (Further discussion on this is in Section 4).

**Comment** The standard decision theory methods [18]-[21] used in screening [7][9] cannot be used to solve the above problem. In standard POMDPs, the interval between two decision epochs (in this case, screening times) is fixed exogenously; in standard POSMDPs, the time between two decision epochs is the sojourn time for the underlying core-state process. In our setting, the time between two decision epochs depends on the action (follow-up date), the external information process, and the state trajectory. In standard POMDPs (POSMDPs) the cost incurred in a decision epoch depend on the current state, while in the above problem the delay cost depends on the state trajectory. Moreover, in our setting the disease state trajectory is not restricted to a Markovian or Semi-Markovian process.

## 3 Proposed Approach

**Beliefs** By a *belief $b$* we mean a probability distribution over the pairs consisting of state trajectories and a label $l$ for the diagnosis: $l = 1$ if the patient has been diagnosed with the disease, $l = 0$ otherwise. By definition, a belief is a function $b : \mathcal{S}^T \times \{0, 1\} \to [0, 1]$ such that $\sum_{\vec{s}, l} b(\vec{s}, l) = 1$ but it is often convenient to view a belief as a vector. Beliefs are updated using Bayesian updating every time there is a new observation (test outcomes, patient arrival, external information). Knowledge of beliefs will be sufficient to solve the optimization problem (1); see the Appendix C in the Supplementary Materials. We write $\mathcal{B}$ for the space of all beliefs.

**Bellman Equations** To solve (1) we will formulate and solve the Bellman equations. To this end, we begin by defining the various components of the Bellman equations. Fix a time $t$. The cost $\tilde{C}$ incurred at time $t$ depends on what happens at that time: i) if the patient (with diagnosis status $l = 0$ before the test) is tested and found to have acquired the disease, the cost is the sum of the cost of testing and the cost of delay, ii) if the patient has the disease and is not detected, then the cost of delay is incurred in the time slot $T$, and iii) if the patient does not have the disease, then the cost incurred in time slot $t$ depends on whether a test was done in time slot $t$ or not. We write these cases below.

$$\tilde{C}(\vec{s}, t, z, l) = \begin{cases} wC(t - t_D; t_D) + (1 - w)\delta^t I(z \neq \emptyset) & t \leq T, l = 0, z \in \mathcal{Z}^+ \\ wC(T - t_D; t_D) & t = T, l = 0 \\ (1 - w)\delta^t I(z \neq \emptyset) & \text{otherwise} \end{cases} \tag{2}$$

A *recommendation plan* $\tau : \mathcal{Z} \to \mathcal{T}$ maps the observation $z$ at the end of time slot $t$ to the next scheduled follow-up time. Note that the recommendation plan is defined for a time $t$ and is different than the policy. Denote the probability distribution over the observations (test outcome $z$, duration to the next arrival $\tilde{\tau}$, and the external information at the next arrival time $y$) conditional on the current belief $b$ and the current recommendation plan $\tau$ by $Pr(z, y, \tilde{\tau}|b, \tau, x)$. The belief $b$ is updated to

$\hat{\boldsymbol{b}}$ in the next arrival time $\tilde{\tau}$ based on the observations, current recommended plan and the current beliefs using Bayesian updating as $\hat{b}(\vec{s}, l) = Pr(\vec{s}, l | \boldsymbol{b}, \boldsymbol{\tau}, y, z, \tilde{\tau}, x)$.

The optimal values for the objective in (2) starting from different initial beliefs can be expressed in terms of a value function $V : \mathcal{B} \times \{1, ..., T+1\} \to \mathbb{R}$. The value function at time $t$ when the patient is screened solves the Bellman equation:

$$V(\boldsymbol{b}, t) = \max_{\boldsymbol{\tau}} \left[ \sum_{\vec{s}, l, z} -b(\vec{s}, l) Pr(z | \vec{s}, x) \left[ \tilde{C}(\vec{s}, t, z, l) \right] + \sum_{z, \tilde{\tau}, y} Pr(z, y, \tilde{\tau} | \boldsymbol{b}, \boldsymbol{\tau}, x) V(\hat{\boldsymbol{b}}, t + \tilde{\tau}) \right] \quad (3)$$

We define $V(\boldsymbol{b}, T+1) = 0$ for all beliefs. Note that the computation of the first term in the RHS of (3) has a worst case computation time of $|\mathcal{S}|^T$. Therefore, solving for exact $V(\boldsymbol{b}, T)$ that satisfies (3) is computationally intractable when $T$ is large. Next, we derive a useful property of the value function. (The proof of this and all other results are in the Appendix D-F of the Supplementary Material.).

**Lemma 1** For every $t$, the value function $V(\boldsymbol{b}, t)$ is the maximum of a finite family of functions that are linear in the beliefs $\boldsymbol{b}$. In particular, the value function is convex and piecewise linear.

The above property was shown for POMDPs in [39], we use the same ideas to extend it to our setup.

### 3.1 Constructing the Exactly Optimal Policy

Every linear function of beliefs is of the form $\boldsymbol{\alpha}^* \boldsymbol{b}$ for some vector $\boldsymbol{\alpha}$. (We view $\boldsymbol{\alpha}, \boldsymbol{b}$ as column vectors and write $\boldsymbol{\alpha}^*$ for the transpose.) Hence Lemma 1 tells us that there is a finite set of vectors $\Gamma(t)$ such that $V(\boldsymbol{b}, t) = \max_{\boldsymbol{\alpha} \in \Gamma(t)} \boldsymbol{\alpha}^* \boldsymbol{b}$. We refer to $\Gamma(t)$ as the *set of alpha vectors*. In view of Lemma 1, to determine the value functions we need only determine the sets of alpha vectors. If we substitute the expression $V(\boldsymbol{b}, t) = \max_{\boldsymbol{\alpha} \in \Gamma(t)} \boldsymbol{\alpha}^* \boldsymbol{b}$ into (3), then we obtain a recursive expression for $\Gamma(t)$ in terms of $\Gamma(t+1)$. By definition, the value function at time $T+1$ is identically 0 so $\Gamma(T+1) = \{\boldsymbol{0}\}$, where $\boldsymbol{0}$ is the $|\mathcal{S}^T \times \{0, 1\}|$ dimensional zero vector, so we have an explicit starting point for this recursive procedure. There is an optimal action associated with each alpha vector. The action corresponding to the optimal alpha vector at a certain belief is the output of the optimal action given that belief, and so constructing the sets of alpha vectors yields the optimal policy; the details of the algorithm are in the Algorithm 3 in the Appendix A of the Supplementary Materials. Unfortunately, the algorithm to compute the sets of alpha vectors is computationally intractable (as expected). We therefore propose an algorithm that is tractable to compute an approximately optimal policy.

### 3.2 Constructing the Approximately Optimal Policy

Point-Based Value Iteration (PBVI) approximation algorithms are known to work well for standard POMDPs [18]. These algorithms rely on choosing a finite set of belief vectors and constructing alpha vectors for these belief vectors and their success depends very much on the efficient construction of the set of belief vectors. The standard approaches [18] for belief construction are not designed to cope with settings like ours when beliefs lie in a very high dimensional space; in our setup belief has $|\mathcal{S}^T \times \{0, 1\}|$ dimensions. In Algorithm 1 (pseudo-code in the Appendix A of the Supplementary Materials), we first construct a lower dimensional belief space by sampling trajectories that are more likely to occur for the disease and then sampling the set of beliefs in the lower dimensional space that are likely to occur over the course of various screening policies. The key steps for Algorithm 1 are

**1. Sample typical physiological state trajectories** Sample a set $\tilde{\mathcal{S}} \subset \mathcal{S}^T$ of $K$ physiological trajectories from the distribution $Pr(\vec{s} | x)$.

**2. Construct the set of reachable belief vectors** Say that a belief vector $\boldsymbol{b}_2$ is *reachable* from the belief vector $\boldsymbol{b}_1$ if it can be derived by Bayesian updating on the basis of some underlying screening policy. We construct the sets of belief vectors that can be reached under different screening policies. For the first time slot, we start with a belief vector that lies in the space $\tilde{\mathcal{S}} \times \{0, 1\}$ given as $Pr(\vec{s} | x) / Pr(\tilde{\mathcal{S}} | x), \ \forall \vec{s} \in \tilde{\mathcal{S}}, l = 0$. For subsequent times, we select the beliefs that are encountered under random exploration of the actions (recommendation of future test dates). In addition to using random exploration, we can choose actions determined from a set of policies such as the clinical policies used in practice [27] [28] [47] to construct the set of reachable belief vectors.

Denote the set of belief vectors constructed at time $t$ by $\bar{B}[t]$ and the set of all such beliefs as $\bar{B} = \{\bar{B}[t], \forall t\}$. We carry out point-based value backups on these beliefs $\bar{B}$ (see Algorithm 2 in the Appendix A of the Supplementary Materials), to construct the alpha vectors and thus the approximately optimal policy. Henceforth, we refer to our approach (Algorithm 1 and 2) as DPSCREEN.

**Computational Complexity**  The worst case computation of the policy requires $\mathcal{O}\left(T(B)^2 T^2 K |\mathcal{Y}||\mathcal{Z}|\right)$ steps, where $B = \max_t |\bar{B}[t]||$ is the maximum over the number of points sampled by the Algorithm 1 for any time slot $t$. The complexity can be reduced by restricting the space of actions; e.g. by bounding the amount of time allowed between successive screenings. Moreover, the proposed algorithms can be easily parallelized (many operations carried inside the iterations in Algorithm 2 can be done parallel), thus significantly reducing computation time.

**Approximation Error**  Because we only sample a finite number of trajectories, the policy we construct is not optimal but we can bound the loss of performance in comparison to the exactly optimal policy and hence justify the term "approximately optimal policy." Define the *approximation error* to be the difference between the value achieved by the exact optimal policy (solution to (1)) and the value achieved by the approximately optimal policy (output from Algorithm 2). As a measure of the density of sampling of the belief simplex we set $\Omega(\bar{B}) = \zeta \max_{t \in \mathcal{T}} \max_{\mathcal{B}} \min_{\boldsymbol{b} \in \bar{B}[t]} ||\boldsymbol{b} - \boldsymbol{b}'||_1$, where $\zeta$ is a constant that measures the maximum expected loss that can occur in one time slot. We make a few assumptions for the proposition to follow. The cost for delay is $C(t_d - t_D; t_D) = c(t_d - t_D)\delta^{t_D}$, where $c(d)$ is a convex function of $d$. The test outcome is accurate, i.e. no false positives and no false negatives. The maximum screening interval is bounded by $W < T$. The time horizon $T$ is sufficiently large. We show that the loss of performance is bounded by the sampling density.

**Proposition 1**  The approximation error is bounded above by $\Omega(\bar{B})$.

### 3.3  Robustness

**Estimation Error**  To this point, it has been assumed that the model parameters are known. In practice, the model parameters need to be estimated using the observational data. In the next section, we will give a concrete example of how we estimate these parameters using observational data for breast cancer. Here we discuss the effect of error in estimation. Suppose that the model being estimated (true model) is $\boldsymbol{m}' \in \boldsymbol{M}$, where $\boldsymbol{M}$ is the space of all the possible models (model parametrizations) under consideration. (We assume that the probability distribution of the physiological state transition, the patient's self-observation outcomes, and the clinician's observation outcomes are continuous on $\boldsymbol{M}$.) Write $\boldsymbol{L} = \boldsymbol{M} \times \mathcal{B}$ for the joint space of models and beliefs. Let the estimate of the model be $\hat{\boldsymbol{m}}$. Let us assume that for every model in $\boldsymbol{M}$ the solution to (1) is unique. Therefore, we can define a mapping $\boldsymbol{\tau}^* : \boldsymbol{L} \times \mathcal{Z} \times \mathcal{T} \to \mathcal{T}^{|\mathcal{Z}|}$, where $\boldsymbol{\tau}^*(l, z, t)$ is the optimal recommended screening time at $l$, at time $t$ following $z$. For a fixed model $\boldsymbol{m}$, $\boldsymbol{\tau}^*((\boldsymbol{m}, \boldsymbol{b}), z, t)$ is the maximizer in (3).

**Theorem 1.**  There is a closed lower dimensional set $\boldsymbol{E} \subset \boldsymbol{L}$ such that the function $\boldsymbol{\tau}^*$ is locally constant on the complement of $\boldsymbol{E}$.

Theorem 1 implies that, with probability 1, if the model estimate $\hat{\boldsymbol{m}}$ and the true model $\boldsymbol{m}'$ are sufficiently close, then the actions recommended by the exactly optimal policies for both models are identical. Therefore, the impact of estimation error on the exactly optimal policy is minimal. However, we construct approximately optimal policies. We can combine these conditions with Proposition 1 to say that if the approximation error $\Omega(\bar{B})$ goes to zero, then the approximately optimal policy (for $\hat{\boldsymbol{m}}$) will also converge to the exactly optimal policy for true model $\boldsymbol{m}'$.

**Personalization:**  Figure 1 provides a graphical representation of the way in which DPSCREEN is personalized to the patients. We consider three Patients. The disease model for each patient is given by the *ex ante* survival curve (the probability of not becoming diseased by a given time). As shown in the graphs, the survival curves for Patients 1, 2 are the same; the survival curve for Patient 3 begins below the survival curve for Patients 1, 2 but is flatter and so eventually crosses the survival curve for Patients 1, 2. All three patients are screened at date 1; for all three the test outcome is $z = \text{Low}$. Hence the belief (risk assessment) for all three patients decreases. As a result, Patients 1, 2 are scheduled for next screening at date 4 but Patient 3, who has a lower *ex ante* survival probability, is scheduled for next screening at date 3. Thus, the policy is *personalized to the ex ante risk*. However, at date 2, all three patients experience an external information shock which causes them to be screened early. The test outcome for Patient 1 is $z = \text{Medium}$ so Patient 1 is assessed to be at higher risk and is scheduled for next screening at date 3; the test outcome for Patient 2 is

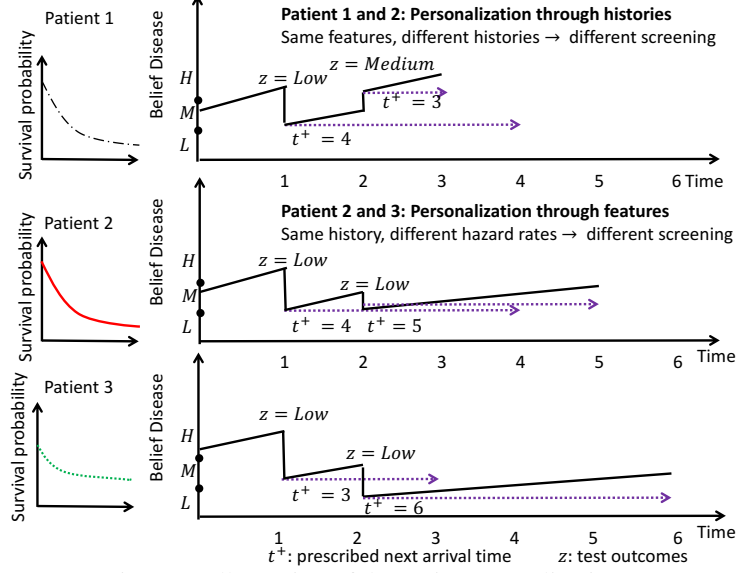

Figure 1: Illustration of dynamic personalization

$z = $ Low so Patient 2 is assessed to be at lower risk and is scheduled for next screening at date 5. Thus the policy is *personalized to the dynamic history*. The test outcome for Patient 3 is $z = $ Low and Patient 3's *ex ante* survival probability is higher so Patient 3's risk is assessed to be very low, and Patient 3 is scheduled for next screening at date 6. Thus the policy adjusts to time-varying model parameters.

## 4 Illustrative Experiments

Here we demonstrate the effectiveness of our policy in a real setting: screening for breast cancer.

**Description of the dataset:** We use a de-identified dataset (from Athena Health Network [22]) of $45,000$ patients aged 60-65 who underwent screening for breast cancer. For most individuals we have the following associated features: age, the number of family members with breast cancer, weight, etc. Each patient had at least one mammogram; some had several. (In total, there are 84,000 mammograms in the dataset.) If the patients had a positive mammogram, a biopsy is carried out. Further description of mammogram output is in the Appendix G of the Supplementary Materials.

**Model description** We model the disease progression using a two-state Markov model: $\mathcal{S} = \{H, D\}$ ($H$ = Healthy, $D$ = Disease/Cancer). Given patient features $x$, the initial probability of cancer is $p_{in}(x)$ and the probability of transition from the $H$ to $D$ is $p_{tr}(x)$. The external information $Y$ is the size (perhaps 0) of a breast lump, based on the patient's own self-examination. In view of the universal growth law for tumor described in [23], we model $Y(t) = g(t) + \epsilon(t)$, where $g(t) = (1 - e^{-\iota(t-t_s)})I(t > t_s)$ is the size of the tumor and $t_s$ is the time at which patient actually develops cancer (the lump exists), $\epsilon(t)$ is a zero mean white noise process with variance $\sigma^2$ and $I()$ is the indicator function. If the lump size $Y$ exceeds the threshold $\tilde{y}$, then the patient visits the clinician, where tests are carried out. The set of test outcomes is $\mathcal{Z} = \{\emptyset, 1, 2, 3\}$, where $z = \emptyset$ when no test is done, $z = 1$ when the mammogram is negative and no biopsy is done, $z = 2$ when the mammogram is positive and the biopsy is negative, $z = 3$ when both mammogram and biopsy is positive.

**Model Estimation** We use the specificity and sensitivity for the mammogram from [7]. Each patient has a different (initial) risk for developing cancer; we compute the risk scores using the Gail model [24], which we use as the feature $x$. We assumed $p_{in}(x)$ and $p_{tran}(x)$ are logistic functions of $x$. We use standard Markov Chain Monte Carlo methods to estimate these functions $p_{in}(x)$ and $p_{tran}(x)$ (further details in the Appendix G of the Supplementary Materials). We assume that each woman has one self-examination per month [25] [26]. We use the value $\iota = 0.9$ as stated in [23]. We estimate the parameters for the self-examinations $\sigma = 0.43$ and $\tilde{y} = 1$ on the basis of the values of sensitivity and specificity for the self-examination from the literature [43]. In the comparisons to follow, we

will also analyze the setting when there are no self-examinations. We divide the population into two risk groups; the Low risk group consists of patients whose prior estimated risk of developing cancer within five years is less than 5%; the High risk group consists of patients whose prior estimated risk exceeds 5%.

**Performance Metrics, Objective and Benchmarks:** Our objective is to minimize the number of screenings subject to a constraint on expected delay cost. We assume the delay cost is linear: $C(t_d - t_D, t_D) = t_d - t_D$. To derive the solution to this constrained problem from construction, which minimizes the weighted sum of screening cost and delay cost, we solve the weighted problem for some weight $w$, and then tune $w$ to select the policy that minimizes the number of screenings subject to a constraint on expected delay cost. For comparison purposes, we take the constraint on expected delay cost to be the expected delay that arises from current clinical practice (annual screening in the US [27][28], biennial screening in some other countries [29]). (Because our objective is to minimize the *number* of screenings, we take the cost of each screening to be 1, whether or not a biopsy is performed.)

**Comment** At this point, we remind that existing frameworks [7][9][10] cannot be used to solve for the optimal screening policy in the above setup because: i) the costs incurred (delay) depends on the state trajectory and not just the current state, and ii) the lump growth model and the patient's self-examination of the lump is not easy to incorporate in these works.

**Comparisons with clinical screening policies:** We compare our constructed policies (for the two groups), with and without self-examination, in terms of three metrics: i) $E[N|R]$: the expected number of tests per year, conditional on the risk group; ii) $E[\Delta|R]$: the expected delay, conditional on the risk group; iii) $E[\Delta|R, D]$: the expected delay, conditional on the risk group and the patient actually developing cancer. Because $E[\Delta|R]$ is the expected *unconditional delay*, it accounts for patients who do not develop cancer as well as for patients who do have cancer; because most patients do not develop cancer, $E[\Delta|R]$ is small. We show the comparisons with the annual policies in Table 1; we show the comparisons with biennial screening in the Appendix G of the Supplementary Materials.

In Table 1 we compare the performance of DPSCREEN (with and without self-examination) for Low and High risk groups against the current clinical policy of annual screening. For both risk groups, the proposed policy achieves approximately the same expected delay as the benchmark policy while doing many fewer tests (in expectation). With self-examinations, the expected reduction in number of screens is 57-68% (depending on risk group); even without self-examinations, the expected reduction in number of screens is 28-45% percent (depending on risk group).

In Table 2 we contrast the difference in DPSCREEN across the two risk groups. To keep the comparison fair, we fix the tolerance in the delay to a fixed value. The proposed policy is personalized as it recommends significantly fewer tests to the low risk patients in contrast to the high risk patients.

**Impact of the type of disease:** We have so far considered breast cancer as an example and assumed linear delay costs. For some diseases (such as Pancreatic cancer [30][5]) the survival probability decreases very quickly with the delay in detection and therefore it might be reasonable to assume a cost of delay that is strictly convex (such as quadratic costs) in delay time for some disease. In Figure 2, we show that for a fixed risk group and for the same weights the policy constructed using quadratic costs is much more aggressive in testing. Moreover, the regime of operation of the policy (the points achieved by the policy in the 2-D plane

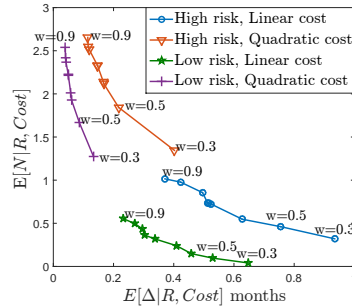

Figure 2: Impact of the type of disease

$E[N|R, \text{Cost}]$ vs $E[\Delta|R, \text{Cost}]$) can vary a lot depending on the choice of cost function even though the same weights are used. Therefore, the cost should be chosen based on the disease.

## 5 Related Works

In Section 2, following the equation (1), we compared our methods with frameworks to some general frameworks in decision theory [18]-[21]. Next, we compare with other relevant works.

Table 1: Comparison of the proposed policy with annual screening for both high and low risk group.

| Risk Group | Metrics | DPSCREEN with self-examination | DPSCREEN w/o self-examination | Annual |
|---|---|---|---|---|
| Low | $E[N|R], E[\Delta|R], E[\Delta|R, \mathrm{D}]$ | 0.32, 0.23, 9.2 | 0.55, 0.23, 9.2 | 1, 0.24, 9.6 |
| High | $E[N|R], E[\Delta|R], E[\Delta|R, \mathrm{D}]$ | 0.43, 0.50, 6.7 | 0.72, 0.52, 7.07 | 1, 0.52, 7.07 |

Table 2: Comparison of the proposed policies across different risk groups

| Risk Group | DPSCREEN with self-examination $E[N|R], E[\Delta|R], E[\Delta|R, \mathrm{D}]$ | DPSCREEN w/o self-examination $E[N|R], E[\Delta|R], E[\Delta|R, \mathrm{D}]$ |
|---|---|---|
| Low | 0.12, 0.33, 13.7 | 0.32, 0.33, 13.7 |
| High | 0.80, 0.35, 4.73 | 1.09, 0.35, 4.73 |

**Screening frameworks for different diseases in operations research:** Many works have focused on optimizing population-based screening schedules, which are not personalized (See [42] and references therein). In [7] [9] the authors develop personalized POMDP based screening models. The underlying disease evolution (breast and colon cancer) is assumed to follow a Markov process. External information process such as self-exams and the test outcomes over time are assumed to follow a stationary i.i.d process given the disease process. In [10] authors develop personalized screening models based on principles of Bayesian design for maximizing information gain (based on [40]). The underlying disease model (cardiac disease) is a dynamic (two-state) survival model and the cost of misdetection is a constant and does not depend on the delay. The test outcomes are modeled using generalized linear mixed effects models, and there is no external information process. To summarize, all of the above methods rely on very specific models for their disease, test outcomes, and external information, while our method imposes much less restrictions on the same.

**Screening frameworks for different diseases in medical literature:** The Medical research literature on screening (e.g., Cancer Intervention and Surveillance Modelling Network, US preventive services task force, etc.) relies on stochastic simulation based methods: fix a disease model and a set of screening policies to be compared; for each policy in the set, simulate outcome paths from the model; compare across the set of policies [44]-[48]. The clinical guidelines for screening issued by the US preventive services task force [47][49] for colon cancer cancers are created based on the MISCAN-COLON [46] model for colon cancer. Simulations were carried out to compare different screening policies suggested by experts for that specific disease model- MISCAN-COLON. This approach allows more realistically complex models but it only compares a fixed set of policies, all of which may be far from optimal.

**Controlled Sensing:** In controlled sensing [21][34][38] the problem of sensor scheduling requires deciding which sensor to use and when; this problem is similar the personalized screening problem studied here. In these works [21][34][38], the main focus is to exploit (or derive) structural properties of the process being sensed and the cost functions such that the exactly optimal sensing schedule is easy to characterize and compute. The structural assumptions such as the process that is sampled is stationary and Markov make these works less suited for personalized screening.

# 6 Conclusion

In this work, we develop a novel methodology for constructing personalized screening policies that balance the cost of screening against the cost of delay in detection of disease. The disease is modeled as an arbitrary finite state stochastic process with an absorbing disease state. Our method incorporates the possibility of external information, such as self-examination or discovery of co-morbidities, that may trigger arrival of the patient to the clinician in advance of a scheduled screening appointment. We use breast cancer data to develop the disease model. In comparison with current clinical policies, our personalized screening policies reduce the number of screenings performed while maintaining the same delay in detection of disease.

# 7 Acknowledgements

This work was supported by the Office of Naval Research (ONR) and the National Science Foundation (NSF) (Grant number: 1533983 and Grant number: 1407712).

## Footnotes

[1]The restriction to a single absorbing disease state is only for expositional convenience.

[2]For many diseases, it seems natural to identify states intermediate between Healthy and Disease. For instance, because breast lumps [26] or colon polyps [9] that are found to be benign may become malignant, it seems natural to distinguish at least one Risky state, intermediate between the Healthy and Disease states.

[3]We can encompass the possibility of competing risks (e.g., different kinds of heart failure) [13] simply by allowing for multiple absorbing states.

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
