[Supplementary Material · +++++++nips_2017_Screening_jmtd-supp-camera-ready.pdf]

# Supplementary Materials for DPSCREEN: Dynamic Personalized Screening

**Kartik Ahuja**
Electrical and Computer Engineering Department
University of California, Los Angeles
ahujak@ucla.edu

**William R. Zame**
Economics Department
University of California, Los Angeles
zame@econ.ucla.edu

**Mihaela van der Schaar**
Engineering Science Department, University of Oxford
Electrical and Computer Engineering Department, University of California, Los Angeles
mihaela.vanderschaar@oxford-man.ox.ac.uk

## 1 Appendix

### 1.1 Appendix A

In this section, we define the belief update expressions that are required in Algorithm 1. Compute the belief updates $\forall \vec{s}, l \in \tilde{\mathcal{S}} \times \{0,1\}$

$$\Theta(b,z)[\vec{s},l] = \frac{Pr(\vec{s},l,z|x)}{Pr(z|x)} = \frac{\sum_{\tilde{l}} Pr(\vec{s},l,z,\tilde{l}|x)}{\sum_{\vec{s}}\sum_{\tilde{l}} Pr(\vec{s},l,z,\tilde{l}|x)} = \frac{\sum_{\tilde{l}} b(\vec{s},\tilde{l})Pr(z|\vec{s})Pr(l|\tilde{l},z,x)}{\sum_{\vec{s}\in\tilde{\mathcal{S}}}\sum_{\tilde{l}} b(\vec{s},\tilde{l})Pr(z|\vec{s},x)Pr(l|\tilde{l},z,x)} \tag{1}$$

$$\Phi(b,y,\tilde{\tau},\tau,t)[\vec{s},l] = \frac{Pr(\vec{s},l,y,\tilde{\tau}|\tau,t,x)}{\sum_{\vec{s},l} Pr(\vec{s},l,y,\tilde{\tau}|\tau,t,x)} = \frac{Pr(\vec{s},l,y,\tilde{\tau}|\tau,t,x)}{\sum_{\vec{s},l} Pr(\vec{s},l,y,\tilde{\tau}|\tau,t,x)} = \frac{b(\vec{s},l)Pr(y,\tilde{\tau}|\vec{s},\tau,t,x)}{\sum_{\vec{s},l} b(\vec{s},l)Pr(y,\tilde{\tau}|\vec{s},\tau,t,x)} \tag{2}$$

For all $\tilde{\tau} \le \tau$ we have

$$Pr(y,\tilde{\tau}|\vec{s},\tau,t,x) = Pr(\{Y(s) \le \tilde{y};\ \forall s \le t+\tilde{\tau}\}, Y(t+\tilde{\tau}) = y|\vec{s},\tau,t,x) \tag{3}$$

For all $\tilde{\tau} \ge \tau$ we have

$$Pr(y,\tilde{\tau}|\vec{s},\tau,t,x) = 0 \tag{4}$$

$$\max_{\boldsymbol{\tau}} \Big[ \sum_{\vec{s},l,z} b(\vec{s},l)Pr(z|\vec{s})\big[\tilde{C}(\vec{s},t,z)\big] + \delta \sum_{z,\tilde{\tau},y} \max_{\alpha} \sum_{\vec{s},l,\tilde{l}} b(\vec{s},l)\alpha[\vec{s},\tilde{l}]Pr(z|\vec{s},x)Pr(\tilde{l}|l,z,x)Pr(y,\tilde{\tau}|\vec{s},\boldsymbol{\tau}[z],x) \Big] \tag{5}$$

$$\max_{\boldsymbol{\tau}} \Big[ \sum_{\vec{s},l,z} b(\vec{s},l)Pr(z|\vec{s})\Big[\tilde{C}(\vec{s},t,z) + \delta \sum_{\tilde{\tau},y,\tilde{l}} \alpha^*[\vec{s},\tilde{l}]Pr(\tilde{l}|l,z,x)Pr(y,\tilde{\tau}|\vec{s},\boldsymbol{\tau}[z],x) \Big] \Big] \tag{6}$$

---
**Algorithm 1** Constructing the Belief Sets
---
$r = \mathbf{0}_T$ a $T$ dimensional zero vector, $\bar{B}$ is the array to store the belief vectors that can be achieved at the $T$ time instances

Sample $K$ iid samples of the trajectory $Pr(\vec{s}|x)$ to form the set $\tilde{S}$

For each $\vec{s} \in \tilde{S}$

$\bar{B}[1,0](\vec{s}, l = 0) = Pr(\vec{s})/Pr(\tilde{S})$

End

For each $\vec{s} \in \tilde{S}$

    For $t = 1 : T$

        Sample $a \sim \text{Bernoulli}(p)$ (if the patient arrives in that time slot $a = 1$ or not $a = 0$)

        Sample $z \sim Pr(z|\vec{s}, x, a)$, If $a = 0$ (patient does not arrive), then $z = \emptyset$

        $\bar{B}[j,t] = \Theta(\bar{B}[j, t-1], z, t)$ (See equation (1))

        Sample $\tau \sim \text{Multi}[1, ..., T - t - 1]$

        Sample $y, \tilde{\tau} \sim Pr(y, \tilde{\tau}|\vec{s}, t, \tau, x)$ (See equation (3) and equation (4))

        $\hat{B} = \Phi(\bar{B}[j,t], y, \tilde{\tau}, \tau, t)$ (See equation (2))

        $\bar{B}[j + K + r(t + \tilde{\tau}), t + \tilde{\tau}] = \hat{B}$

        $r(t + \tilde{\tau}) = r(t + \tilde{\tau}) + 1$

    End

$j = j + 1$

End

Copy belief vectors at time $t$ to the belief at $t + 1$.
---

**Algorithm 2** Approximate Policy Computation Part

---

**Function: TPBVI**
FUNCTION INPUT: Sets $\Gamma(q), \forall q > t$,
For each $\tau \in \{1, ..., T - t - 1\}$
For each $z \in \mathcal{Z}$
    For each $(\tilde{\tau}, y) \in \{t + 1, .., t + \tau\} \times \mathcal{Y}$
        For each $\boldsymbol{\alpha} \in \Gamma(t + \tilde{\tau})$
           For each $\vec{s}, l \in \tilde{\mathcal{S}} \times \{0, 1\}$
           $\boldsymbol{\theta}[\vec{s}, l] = \sum_{\tilde{l}} \boldsymbol{\alpha}[\vec{s}, l] Pr(\tilde{l}|l, z) Pr(y, \tilde{\tau}|\vec{s}, \tau, x)$
           $\boldsymbol{\theta}[\vec{s}, l] = \boldsymbol{\theta}[\vec{s}, l] Pr(z|s, \vec{x})$
           End
           $\Gamma(t, z, \tau, \tilde{\tau}, y) = \Gamma(t, z, \tau, \tilde{\tau}, y) \cup \{\boldsymbol{\theta}\}$
        End
    End
End
End
For each belief point $\boldsymbol{b} \in \bar{B}[, t]$
For each $z \in \mathcal{Z}$
    For each $\tau \in \{1, ..., T - t - 1\}$
    $\boldsymbol{\zeta} = \sum_y \sum_{\tilde{\tau}} \arg\max_{\boldsymbol{\alpha} \in \Gamma(t, z, \tau, \tilde{\tau}, y)} [\boldsymbol{\alpha}]' \boldsymbol{b}$
    $\Gamma(t, z, \tau) = \Gamma(t, z, \tau) \cup \boldsymbol{\zeta}$
    End
End
End
For each belief point $\boldsymbol{b} \in \bar{B}[, t]$
For each $z \in \mathcal{Z}$
    $\{\boldsymbol{\alpha}', (\tau)'\} = \arg\max_{\tau, \boldsymbol{\alpha} \in \Gamma(t, z, \tau)} \sum_{\vec{s}, l} \left[ -\tilde{C}(\vec{s}, t, z, l) + \boldsymbol{\alpha}[\vec{s}, l]) \right] b(\vec{s}, l)$
    $\Gamma(t, z) = \Gamma(t, z) \cup \boldsymbol{\alpha}'$
    $A(t, z) = A(t, z) \cup (\tau)'$
End
$\Gamma(t) = \Gamma(t) + \Gamma(t, z)$
End
FUNCTION OUTPUT: $\Gamma(t), \{\Gamma(t, z), \forall z\}, \{A(t, z), \forall z\}$
$\Gamma(t, z)$ is set of alpha vectors and each one of them is optimal for one of the beliefs in the set $\bar{B}(, t)$, $A(t, z)$ is the set of optimal actions corresponding to the alpha vectors in $\Gamma(t, z)$
For any belief $\boldsymbol{b}$, $z$ find the nearest point in $\bar{B}[, t]$ and use the corresponding alpha vector in $\Gamma(t, z)$ and the corresponding action $A(t, z)$
End
$\Gamma(T + 1) = \{\boldsymbol{0}\}$
For each $t = 0$ to $T - 1$
$\Gamma(T - t) = \text{TPBVI}(\{\Gamma(T - r)\}_{-1 \leq r \leq t-1})$
End

---

**Algorithm 3** Exact Policy Computation Part

NOTATION: $\oplus$ is the Cartesian sum

FUNCTION EVI

FUNCTION INPUT: Sets $\Gamma(q), \forall q > t,, \Gamma(T+1) = \{\mathbf{0}\}$,

For each $z \in \mathcal{Z}$

For each $\tau \in \{1, ..., T - t - 1\}$

    For each $(\tilde{\tau}, y) \in \{t+1, .., t+\tau\} \times \mathcal{Y}$

        For each $\boldsymbol{\alpha} \in \Gamma(t + \tilde{\tau})$

           For each $\vec{s}, l$

           $\boldsymbol{\theta}[\vec{s}, l] = \frac{-1}{|\mathcal{Y}|(\tau - t)|} \tilde{C}(\vec{s}, t, z, l) + \delta \sum_{\tilde{l}} \boldsymbol{\alpha}[\vec{s}, l] Pr(\tilde{l}|l, z, x) Pr(y, \tilde{\tau}|\vec{s}, \tau, x)$

           $\boldsymbol{\theta}[\vec{s}, l] = \boldsymbol{\theta}[\vec{s}, l] Pr(z|\vec{s}, x)$

           End

        $\Gamma(t, z, \tau, \tilde{\tau}, y) = \Gamma(t, z, \tau, \tilde{\tau}, y) \cup \{\boldsymbol{\theta}\}$

        End

    $\Gamma(t, z, \tau) = \text{prune}\left( \Gamma(t, z, \tau) \oplus \Gamma(t, z, \tau, \tilde{\tau}, y) \right)$

    End

$\Gamma(t, z) = \text{prune}\left( \Gamma(t, z) \oplus \Gamma(t, z, \tau) \right)$

End

$\Gamma(t) = \text{prune}\left( \Gamma(t) \oplus \Gamma(t, z) \right)$

End

OUTPUT: $\Gamma(t), \{\Gamma(t, z), \forall z \in \mathcal{Z}\}$,

For optimal action at belief $b$ at time $t$ following observation $z$ choose the optimal alpha vector from $\Gamma(t, z)$ and choose the action corresponding to the alpha vector selected

$\Gamma(T+1) = \{\mathbf{0}\}$

For each $t = 0$ to $T - 1$

$\Gamma(T - t) = \text{EVI}(\{\Gamma(T - r)\}_{-1 \le r \le t-1})$

End

In Algorithm 3, the prune function is taken from [21].

$\vec{S}(t)$ State trajectory through time $t$

$Y(t)$ External information through time $t$

$Z(t)$ Test outcome at time $t$

$\Pi(t)$ Policy recommendation at time $t$

$L(t)$ Diagnosis label at time $t$

$V(t)$ Visit indicator at time $t$

Figure 1: Graphical model for the screening setting

## 1.2 Appendix B

**Graphical Model.**

We define a random variable $V(t)$, where $V(t) = 1$ indicates that the patient visits the clinician in time slot $t$ and is zero otherwise. We define the realization of the visit trajectory as $\vec{v} = [v(1), ..., v(T)]$. Let the screening policy be $\pi$. Next, we define the joint distribution of all the random variables that appear in the model.

The joint distribution of the state trajectory $\vec{s}$, the external information process trajectory $\vec{y}$, the visit trajectory $\vec{v}$, the test outcome trajectory $\vec{z}$ is given as

$$
\begin{aligned}
Pr\left(\vec{s}, \vec{y}, \vec{v}, \vec{z} \middle| x\right) = & Pr\left(\vec{s} \middle| x\right) Pr\left(\vec{y} \middle| \vec{s}, x\right) Pr\left(\vec{v}, \vec{z} \middle| \vec{s}, \vec{y}, x\right) \\
& Pr\left(\vec{s} \middle| x\right) \Pi_t Pr\left(y(t) \middle| \vec{s}(t), x\right) \Pi_t Pr\left(v(t) \middle| \vec{v}(t-1), \vec{z}(t-1), \vec{y}, \vec{s}, \pi\right) Pr\left(z(t) \middle| v(t), \vec{s}\right)
\end{aligned}
$$
(7)

We simplify $Pr\left(\vec{y} \middle| \vec{s}\right)$ as $Pr\left(\vec{y} \middle| \vec{s}\right) = \Pi_t Pr\left(y(t) \middle| \vec{s}, x\right)$, where $\Pi$ is the product operator and $Pr\left(y(t) \middle| \vec{s}, x\right)$ is the probability of $Y(t) = y(t)$ conditional on the entire state trajectory. We assumed that the observations $y(t)$ conditional on state trajectory through $\vec{s}(t)$ is independent of other random variables in the model. Therefore, $Pr\left(y(t) \middle| \vec{s}, x\right) = Pr\left(y(t) \middle| \vec{s}(t), x\right)$.

We simplify $Pr\left(\vec{v}, \vec{z} \middle| \vec{s}, \vec{y}, x\right)$ as $Pr\left(\vec{v}, \vec{z} \middle| \vec{s}, \vec{y}, x\right) = \Pi_t Pr\left(v(t) \middle| \vec{v}(t-1), \vec{z}(t-1), \vec{y}, \vec{s}, \pi\right) Pr\left(z(t) \middle| v(t), \vec{s}\right)$ where $Pr\left(v(t) \middle| \vec{v}(t-1), \vec{z}(t-1), \vec{y}, \vec{s}, \pi\right)$ is the probability of visit in time $t$ conditional on visit indicator in time $t-1$, the test outcomes through time $t-1$, the entire external information process trajectory, the state trajectory $\vec{s}$ and the policy $\pi$ and $Pr\left(z(t) \middle| v(t), \vec{s}(t)\right)$ is the probability of test outcome conditional on visit and the state trajectory. Note that $z(t)$'s value when there is a visit depends only on the state trajectory through time $t$. If there is no visit, then $z(t) = \emptyset$. It is easy to simplify $Pr\left(v(t) \middle| \vec{v}(t-1), \vec{z}(t-1), \vec{y}, \vec{s}, \pi\right)$. Based on all the observations until time $t-1$ the policy $\pi$ would have recommended a next screening time. If the next screening time is not $t$, then $Pr\left(v(t) = 1 \middle| \vec{v}(t-1), \vec{z}(t-1), \vec{y}, \vec{s}, \pi\right) = Pr\left(Y(t) > \tilde{y} \middle| \vec{s}, x\right)$. If the screening time is $t$, then $Pr\left(v(t) = 1 \middle| \vec{v}(t-1), \vec{z}(t-1), \vec{y}, \vec{s}, \pi\right) = 1$

## 1.3 Appendix C

**Sufficient statistic for the history:** In this section, our aim is to show that instead of considering the entire history the clinician can only use the belief that we constructed in Section 3.

The history through time $t$ when the patient arrives can be written as $h(t) = \Big[ z(t), y(t), \{y(r) \leq \tilde{y}\}_{r=t-\tilde{\tau}}^{t-1}, \tilde{\tau}, \tau(t-\tilde{\tau}), h(t-\tilde{\tau}) \Big]$, where $z(t)$ is the test outcome at time $t$, $y(t)$ is the external observation at time $t$, and $\tau(t - \tilde{\tau})$ was the prescribed arrival time in the last arrival which occurred at time $t - \tilde{\tau}$.

Write the probability that the patient's state trajectory is $\vec{s}$, the diagnosis state $l$ (the diagnosis state corresponds to the state after the observation $z(t)$ in time slot $t$), conditioned on the history $h(t)$ as $Pr(\vec{s}, l | h(t))$. Next, we describe how to compute $Pr(\vec{s}, l | h(t))$ in terms of the probability distribution $Pr(\vec{s}, \tilde{l} | h(t - \tilde{\tau}))$.

$$
\begin{aligned}
Pr\Big(\vec{s}, l, v \Big| h(t)\Big) &= \frac{Pr\Big(\vec{s}, l, z(t), y(t), \{y(r) \leq \tilde{y}\}_{r=t-\tilde{\tau}}^{t-1} \Big| \tau(t-\tilde{\tau}), h(t-\tilde{\tau})\Big)}{Pr\Big(z(t), y(t), \{y(r) \leq \tilde{y}\}_{r=t-\tilde{\tau}}^{t-1} \Big| \tau(t-\tilde{\tau}), h(t-\tilde{\tau})\Big)} \\
&= \frac{\sum_{\tilde{l}} Pr\Big(\vec{s}, \tilde{l}, l, z(t), y(t), \{y(r) \leq \tilde{y}\}_{r=t-\tilde{\tau}}^{t-1} \Big| \tau(t-\tilde{\tau}), h(t-\tilde{\tau})\Big)}{Pr\Big(z(t), y(t), \{y(r) \leq \tilde{y}\}_{r=t-\tilde{\tau}}^{t-1} \Big| h(t-\tilde{\tau})\Big)} \\
&= \frac{\sum_{\tilde{l}} Pr\Big(\vec{s}, \tilde{l} \Big| h(t-\tilde{\tau})\Big) Pr\Big(y(t), \{y(r) \leq \tilde{y}\}_{r=t-\tilde{\tau}}^{t-1} \Big| \vec{s}, \tau(t-\tilde{\tau})\Big) Pr\Big(z(t)|\vec{s}\Big) Pr\Big(l|\tilde{l}, z(t)\Big)}{Pr\Big(z(t), y(t) \Big| h(t-\tilde{\tau})\Big)}
\end{aligned}
\tag{8}
$$

In the above equation, $Pr\Big(l|\tilde{l}, z(t)\Big)$ is the probability of the new diagnosis state conditional on the existing diagnosis state. If the existing diagnosis state is 1, then the new diagnosis state has to be 1. If the existing diagnosis state is 0, then the new diagnosis state is 1 if $z \in \mathcal{Z}^+$ and 0 otherwise.

By definition the belief at time $t$ is $Pr(\vec{s}, l | h(t))$, which we write as $\hat{\boldsymbol{b}}$ and we write $Pr(\vec{s}, l | h(t - \tilde{\tau}))$ as $\boldsymbol{b}$.

$$
\hat{b}(\vec{s}, l) = \frac{\sum_{\tilde{l}} b(\vec{s}, l) Pr\Big(y(t), \{y(r) \leq \tilde{y}\}_{r=t-\tilde{\tau}}^{t-1} \Big| \vec{s}, \tau(t-\tilde{\tau})\Big) Pr\Big(z(t) \Big| \vec{s}\Big) Pr\Big(l \Big| \tilde{l}, z(t)\Big)}{Pr\Big(z(t), y(t) \Big| h(t-\tilde{\tau})\Big)}
\tag{9}
$$

From the above equation, we can conclude that keeping a track of beliefs is sufficient as the previous belief can be used to compute the new belief (combined with the distributions over the observations).

## 1.4 Appendix D

**Proof of Lemma 1.**

We re-write the value function defined for time slot $t$, which is also a decision epoch (the patient arrives in this slot and test is done) in equation (3) in the main text below.

$$V(\boldsymbol{b}, t) = \max_{\boldsymbol{\tau}} \left[ -\sum_{\vec{s},l,z} b(\vec{s},l) Pr(z|\vec{s},x) \left[ \tilde{C}(\vec{s},t,z,l) \right] + \sum_{z,\tilde{\tau},y} Pr(z,y,\tilde{\tau}|\boldsymbol{b},\boldsymbol{\tau},x) V\left( \hat{\boldsymbol{b}}, t+\tilde{\tau} \right) \right] \quad (10)$$

where $\hat{b}(\vec{s},l) = Pr\left( \vec{s}, l \middle| \boldsymbol{b}, \boldsymbol{\tau}, y, z, \tilde{\tau}, x \right) = \frac{Pr(\vec{s},l,y,z,\tilde{\tau}|\boldsymbol{b},\boldsymbol{\tau})}{Pr(y,z,\tilde{\tau}|\boldsymbol{b},\boldsymbol{\tau})}$. Note that $l$ in the above equation is the diagnosis state before the test outcome $z$ is observed.

$Pr(\vec{s},l,y,z,\tilde{\tau}|\boldsymbol{b},\boldsymbol{\tau})$ is the probability that the patient's trajectory is $\vec{s}$, the test outcome in time slot $t$ is $z$, the external information on patient's next arrival, which occurs $\tilde{\tau}$ time slots later is $y$ conditioned on the recommendation plan $\boldsymbol{\tau}$. We simplify $Pr(\vec{s},l,y,z,\tilde{\tau}|\boldsymbol{b},\boldsymbol{\tau},t)$ as

$$Pr\left( \vec{s}, l, y, z, \tilde{\tau} \middle| \boldsymbol{b}, \boldsymbol{\tau}, t \right) = \sum_{\tilde{l}} Pr\left( \vec{s}, \tilde{l}, l, y, z, \tilde{\tau} \middle| \boldsymbol{b}, \boldsymbol{\tau}, t \right) = \sum_{\tilde{l}} b(\vec{s},\tilde{l}) Pr\left( l, y, z, \tilde{\tau} \middle| \vec{s}, \tilde{l}, \boldsymbol{\tau}, t \right)$$

We simplify $Pr(l, y, z, \tilde{\tau}|\vec{s}, \tilde{l}, \boldsymbol{\tau}, t)$ as

$$Pr(l, y, z, \tilde{\tau}|\vec{s}, \tilde{l}, \boldsymbol{\tau}, t) = Pr(z|\vec{s}, \tilde{l}, \boldsymbol{\tau}, t) Pr(l|\tilde{l}, z, \boldsymbol{\tau}, t, \vec{s}) Pr(y, \tilde{\tau}|z, \vec{s}, \boldsymbol{\tau}, l, \tilde{l}, t)$$
$$= Pr(z|\vec{s}) Pr(l|\tilde{l}, z) Pr(y, \tilde{\tau}|z, \vec{s}, \boldsymbol{\tau}, t) \quad (11)$$

where $Pr(l|\tilde{l}, z)$ is the transition probability from current diagnosis label $\tilde{l}$ to the new label $l$ following the observation $z$. If the patient is diagnosed to be unhealthy, then the diagnosis label continues to be one. If the patient is not diagnosed, then the label turns to one from zero as soon as the patient is diagnosed. Formally stated, $Pr(l=0|\tilde{l}=0, z) = 0, \forall z \in \mathcal{Z}^+$, $Pr(l=0|\tilde{l}=0, z) = 1, \forall z \in [\mathcal{Z}^+]^c$, $Pr(l=0|\tilde{l}=1, z) = 0, \forall z \in \mathcal{Z}$. In the above equation (11), we wrote $Pr(z|\vec{s}, \tilde{l}, \boldsymbol{\tau}, t) = Pr(z|\vec{s})$; this is true because the test outcome is independent of whether the patient has been diagnosed or not, the recommendation plan and the time. We also state $Pr(l|\tilde{l}, z, \boldsymbol{\tau}, t, \vec{s}) = Pr(l|\tilde{l}, z)$ where we use the condition that $l$ is independent of $\boldsymbol{\tau}, t, \vec{s}$ conditional on $\tilde{l}, z$ (this follows from the definition of $l$). Also, if the state trajectory, the recommended time of next arrival, and the current time are known, then the distribution of external information at next arrival time and next arrival time is completely specified by $Pr(y, \tilde{\tau}|z, \vec{s}, \boldsymbol{\tau}, t)$ and whether the patient has been diagnosed or not does not enter the external information process $Pr(y, \tilde{\tau}|z, \vec{s}, \boldsymbol{\tau}, l, \tilde{l}, t) = Pr(y, \tilde{\tau}|z, \vec{s}, \boldsymbol{\tau}, t)$ (this follows from the definition of external information process).

For all $\tilde{\tau} \leq \boldsymbol{\tau}(z)$ we have

$$Pr(y, \tilde{\tau}|z, \vec{s}, \boldsymbol{\tau}, t) = Pr\left( \left\{ Y(s) \leq \tilde{y};\ t < \forall s \leq t + \tilde{\tau} \right\}, Y(t + \tilde{\tau}) = y \middle| \vec{s} \right) \quad (12)$$

For $\tilde{\tau} > \boldsymbol{\tau}(z)$

$$Pr(y, \tilde{\tau}|z, \vec{s}, \boldsymbol{\tau}, t) = 0 \quad (13)$$

Thus we can write the updated belief as

$$\hat{b}(\vec{s}, \tilde{l}) = \frac{Pr(\vec{s}, \tilde{l}, y, z, \tilde{\tau}|\boldsymbol{b}, \boldsymbol{\tau})}{Pr(y, z, \tilde{\tau}|\boldsymbol{b}, \boldsymbol{\tau})} = \frac{\sum_l b(\vec{s},l) Pr(\tilde{l}, y, z, \tilde{\tau}|\vec{s}, l, \boldsymbol{\tau}, t)}{Pr(y, z, \tilde{\tau}|\boldsymbol{b}, \boldsymbol{\tau})} = \frac{\sum_l b(\vec{s},l) Pr(z|\vec{s}) Pr(\tilde{l}|l, z) Pr(y, \tilde{\tau}|z, \vec{s}, \boldsymbol{\tau}, t)}{Pr(y, z, \tilde{\tau}|\boldsymbol{b}, \boldsymbol{\tau})} \quad (14)$$

In this proof uptil now we have computed the expression for $\hat{\boldsymbol{b}}$.

We will use principle of induction to prove the above result. The claim in the Lemma holds for the value function in time slot $T + 1$ as it is defined to be identically zero. Next, we assume that the condition in the Lemma holds for all $r > t$. Therefore, we can write

$$V(\hat{\boldsymbol{b}}, t+\tilde{\tau}) = \max_{\boldsymbol{\alpha} \in \Gamma(t+\tilde{\tau})} \sum_{\vec{s}, \tilde{l}} \boldsymbol{\alpha}[\vec{s}, \tilde{l}] \hat{b}(\vec{s}, \tilde{l}) = \max_{\boldsymbol{\alpha} \in \Gamma(t+\tilde{\tau})} \sum_{\vec{s}, \tilde{l}, l} \boldsymbol{\alpha}[\vec{s}, \tilde{l}] \frac{b(\vec{s}, l) Pr(z|\vec{s}) Pr(\tilde{l}|l, z) Pr(y, \tilde{\tau}|z, \vec{s}, \boldsymbol{\tau}, t+\tilde{\tau})}{Pr(y, z, \tilde{\tau}|\boldsymbol{b}, \boldsymbol{\tau})} \quad (15)$$

Suppose that each $\boldsymbol{\alpha} \in \Gamma(t + \tilde{t})$ is indexed. Henceforth, we also write the index of $\boldsymbol{\alpha}$ in superscript as well, i.e. $\boldsymbol{\alpha}^k$. Define a function $\tilde{k} : \Delta \times \mathcal{T} \times \mathcal{T} \times \mathcal{Y} \times \mathcal{T} \to \mathbb{Z}$ as follows.

$$\tilde{k}(\boldsymbol{b}, \boldsymbol{\tau}, \tilde{\tau}, y, t, z) = \arg \max_k \sum_{\vec{s}, l, \tilde{l}} \boldsymbol{\alpha}^k [\vec{s}, \tilde{l}] b(\vec{s}, l) Pr(z|\vec{s}) Pr(\tilde{l}|l, z) Pr(y, \tilde{\tau}|z, \vec{s}, \boldsymbol{\tau}, t + \tilde{\tau}) \quad (16)$$

Substituting (15) and (16) into (10) we obtain

$$V(\boldsymbol{b}, t) = \max_{\boldsymbol{\tau}} \left[ \sum_{\vec{s}, l, z} b(\vec{s}, l) Pr(z|\vec{s}) \left[ \tilde{C}(\vec{s}, t, z, l) \right] + \sum_{z, \tilde{\tau}, y} \sum_{\vec{s}, l, \tilde{l}} \boldsymbol{\alpha}^{\tilde{k}}[\vec{s}, l] b(\vec{s}, l) Pr(z|\vec{s}) Pr(\tilde{l}|l, z) Pr(y, \tilde{\tau}|z, \vec{s}, \boldsymbol{\tau}, t + \tilde{\tau}) \right]$$

$$\max_{\boldsymbol{\tau}} \left[ \sum_{\vec{s}, l, z} b(\vec{s}, l) Pr(z|\vec{s}) \left[ \tilde{C}(\vec{s}, t, z, l) + \sum_{\tilde{\tau}, y, \tilde{l}} \boldsymbol{\alpha}^{\tilde{k}}[\vec{s}, \tilde{l}] Pr(\tilde{l}|l, z) Pr(y, \tilde{\tau}|z, \vec{s}, \boldsymbol{\tau}, t + \tilde{\tau}) \right] \right]$$

$$\max_{\boldsymbol{\tau}} \left[ \sum_{\vec{s}, l, z, \tilde{\tau}, y, \tilde{l}} b(\vec{s}, l) Pr(z|\vec{s}) \left[ \tilde{C}(\vec{s}, t, z, l) \frac{1}{\omega} + \boldsymbol{\alpha}^{\tilde{k}}[\vec{s}, l] Pr(\tilde{l}|l, z) Pr(y, \tilde{\tau}|z, \vec{s}, \boldsymbol{\tau}, t + \tilde{\tau}) \right] \right]$$

$$(17)$$

where $\omega$ is the total possible combinations of $\tilde{\tau}$, $y$ and $\tilde{l}$. In the above (17), we only use $\tilde{k}$ instead of the entire function $\tilde{k}(\boldsymbol{b}, \boldsymbol{\tau}, \tilde{\tau}, y, t, z)$ for clearer notation.

Observe that the function $\tilde{k}(\boldsymbol{b}, \boldsymbol{\tau}, \tilde{\tau}, y, t, z)$ can take finitely many values. Therefore, for a fixed combination of values $z, \tilde{\tau}, y$ the space $\mathcal{B}$ is thus partitioned into regions where $\tilde{k}(\boldsymbol{b}, \boldsymbol{\tau}, \tilde{\tau}, y, t, z)$ takes a fixed value. Hence, the term $\left[ \tilde{C}(\vec{s}, t, z, l) \frac{1}{\omega} + \boldsymbol{\alpha}^{\tilde{k}}[\vec{s}, l] Pr(\tilde{l}|l, z) Pr(y, \tilde{\tau}|z, \vec{s}, \boldsymbol{\tau}, t + \tilde{\tau}) \right]$ takes a fixed value in each partition as well and this is true of $\forall \vec{s}, l$. Finally, we can create a common refinement of the partitions such that the $\sum_{z, \tilde{\tau}, y} Pr(z|\vec{s}) \left[ \tilde{C}(\vec{s}, t, z, l) \frac{1}{\omega} + \boldsymbol{\alpha}^{\tilde{k}}[\vec{s}, l] Pr(\tilde{l}|l, z) Pr(y, \tilde{\tau}|z, \vec{s}, \boldsymbol{\tau}, t + \tilde{\tau}) \right]$ is fixed for each partition. Therefore, we have so far that the term inside (17) is piecewise linear. The first term inside (17) is convex (since it is linear). The second term inside (17) is convex because of the definition of (16). Thus, the term inside the max operator (17) is piecewise linear and convex. The maximum of piecewise linear and convex functions is also piecewise linear and convex. This proves the result.

## 1.5 Appendix E

**Proof of Proposition 1.** We first re-write the expression for the cost incurred in a time slot below

$$\tilde{C}(\vec{s}, t, z, l) = \begin{cases} wC(t - t_D; t_D) + (1 - w)\delta^t I(z \neq \emptyset) & t \leq T, l = 0, z \in \mathcal{Z}^+ \\ wC(T - t_D; t_D) & t = T, l = 0 \\ (1 - w)\delta^t I(z \neq \emptyset) & \text{otherwise} \end{cases} \tag{18}$$

We substitute $C(t - t_D; t_D) = c(t - t_D)\delta^{t_D}$ to obtain

$$\tilde{C}(\vec{s}, t, z, l) = \begin{cases} wc(t - t_D)\delta^{t_D} + (1 - w)\delta^t I(z \neq \emptyset) & t \leq T, l = 0, z \in \mathcal{Z}^+ \\ wc(t - t_D)\delta^{t_D} & t = T, l = 0 \\ (1 - w)\delta^t I(z \neq \emptyset) & \text{otherwise} \end{cases} \tag{19}$$

Define another function $\hat{C}$ as follows.

$$\hat{C}(\vec{s}, t, z, l) = \begin{cases} wc(t - t_D)\delta^t + (1 - w)\delta^t I(z \neq \emptyset) & t \leq T, l = 0, z \in \mathcal{Z}^+ \\ wc(t - t_D)\delta^t & t = T, l = 0 \\ (1 - w)\delta^t I(z \neq \emptyset) & \text{otherwise} \end{cases} \tag{20}$$

Next, we derive an upper bound on time to detection $t_d$ in terms of the time of incidence $t_D$. Disease starts at $t_D$ and the next screening has to occur at time at most $t_D + W$. Since there are no false positives and false negatives the patient is detected in the next screening. Therefore, we have $t_D \leq t_d \leq t_D + W$.

We derive an upper bound on the difference between $\hat{C}$ and $\tilde{C}$ as

$$\begin{aligned} \tilde{C}(\vec{s}, t, z, l) - \hat{C}(\vec{s}, t, z, l) &\leq (\delta^{T_D} - \delta^{t_d})(c(t_d - t_D)) \\ &\leq \delta^{T_D}(1 - \delta^W)(c(W)) \\ &\leq (1 - \delta^W)(c(W)) \end{aligned} \tag{21}$$

We require that $\tilde{C}(\vec{s}, t, z, l) - \hat{C}(\vec{s}, t, z, l) \leq \kappa$. It is sufficient to bound $(1 - \delta^W)(c(W)) \leq \kappa \implies \delta \geq (1 - \frac{\kappa}{c(W)})^{1/W}$. Henceforth, we assume that $\delta \geq \delta^* = (1 - \frac{\kappa}{c(W)})^{1/W}$. Therefore, we have

$$\hat{C}(\vec{s}, t, z, l) \leq \tilde{C}(\vec{s}, t, z, l) \leq \hat{C}(\vec{s}, t, z, l) + \kappa \tag{22}$$

$\forall \vec{s}, t, z, l$. It can be shown that the solutions to (**??**) with $\hat{C}$ instead of $\tilde{C}$ only differ by $\kappa$ (at most). Let the optimal policy and the corresponding optimal value when cost is $\tilde{C}$ be given as $\pi_1$ and $C_1(\pi_1)$ $(C_2(\pi_2))$. From (22) we have

$$\begin{aligned} C_2(\pi_1) &\leq C_1(\pi_1) \leq C_2(\pi_1) + \kappa \\ C_2(\pi_2) &\leq C_1(\pi_2) \leq C_2(\pi_2) + \kappa \end{aligned} \tag{23}$$

From the definition of $C_1$ and $C_2$ the following can be derived

$$C_2(\pi_2) \leq C_1(\pi_1) \leq C_2(\pi_1) + \kappa \leq C_2(\pi_2) + \kappa \tag{24}$$

Next, we will use $\hat{C}$ instead of $\tilde{C}$. Define a function $\bar{C}(\vec{s}, t, z, s) = \frac{\hat{C}(\vec{s}, t, z, l)}{\delta^t}$.

We write the value function for the modified objective as

$$\bar{V}(\boldsymbol{b}, t) = \max_{\boldsymbol{\tau}} \left[ \sum_{\vec{s}, l, z} b(\vec{s}, l) Pr(z|\vec{s}) \left[ \bar{C}(\vec{s}, t, z, l) \right] + \delta \sum_{z, \tilde{\tau}, y} Pr(z, y, \tilde{\tau}|\boldsymbol{b}, \boldsymbol{\tau}) \bar{V}(\hat{\boldsymbol{b}}, t + \tilde{\tau}) \right] \tag{25}$$

If $T$ is sufficiently large, then the difference between the value function of the finite horizon and the infinite horizon version of the problem can be made as small as desired.

The maximum difference between the value function computed upto the infinite horizon versus one that is truncated at time $T$ is $\delta^T(1/(1-\delta) + c(W))$. Suppose we want to bound the difference by $\eta$. If $\delta^T(1/(1-\delta) + c(W)) \leq \eta \implies \delta^T \leq \eta(1-\delta) + c(W)\eta$. If $T \geq \max\{\frac{\log(\frac{\eta}{2+\eta})}{\log(\delta)}, \frac{\log(\frac{\eta}{2c(W)})}{\log(\delta)}\}$, then the difference is bounded by $\eta$. Let us consider the infinite horizon for $\bar{V}$ above. We will construct the proof for the infinite horizon version of the problem and then use the above observation to extend the proof to finite horizon.

From equation (25), we can define an operator given as $\Phi_t$ defined as follows.

$$\Phi_t(V) = \max_{\boldsymbol{\tau}} \left[ \sum_{\vec{s},l,z} b(\vec{s},l)Pr(z|\vec{s})\left[\bar{C}(\vec{s},t,z,l)\right] + \delta \sum_{z,\tilde{\tau},y} Pr(z,y,\tilde{\tau}|\boldsymbol{b},\boldsymbol{\tau})V\left(\gamma(\boldsymbol{b},y,z,\boldsymbol{\tau})\right) \right] \quad (26)$$

where $\gamma$ is the belief update operator that can be defined based on the equation (14) in the proof of Lemma 1. Based on standard arguments used to show that a Bellman operator is a contraction mapping [21], we can show that the above operator is a contraction mapping as well with a contraction factor $\delta$.

Similarly, we define an operator $\tilde{\Phi}_t$ associated with our algorithm. Our algorithm takes alpha vectors as input and generates a new set of alpha vectors. Since the set of alpha vectors define the value function (see Lemma 1), we can view the proposed procedure to be an operator that maps a value function to another value function. Define the error introduced by one iteration of the approximate backup $\tilde{\Phi}_t V^B(:,t)$ as $\epsilon = \max_{\boldsymbol{b} \in \Delta} |\tilde{\Phi}_t V^{\bar{B}}(\boldsymbol{b}) - \Phi_t V^{\bar{B}}(\boldsymbol{b})|$. Note that the backup at time $t$ will use $\bar{B}[;t]$ as the input vector of beliefs. Define the density $\delta_{\bar{B}[t]}$ of a set of points $\bar{B}[t]$ to be the maximum distance from any belief in the simplex $\Delta$ to a belief in the set $\bar{B}[t]$.

$$\delta_{\bar{B}[t]} = \max_{\boldsymbol{b}' \in \Delta} \min_{\boldsymbol{b} \in \bar{B}[t]} ||\boldsymbol{b} - \boldsymbol{b}'||_1 \quad (27)$$

We now compute the maximum value $\epsilon$. Let $\boldsymbol{b}' \in \Delta$ be the point where proposed procedure makes the largest error and let $\boldsymbol{b} \in \bar{B}[t]$ be the closest 1-norm sampled belief to $\boldsymbol{b}'$. Let $\boldsymbol{\alpha}$ be the vector maximal at $\boldsymbol{b}$ (this vector is generated by the backup at $\boldsymbol{b}$ because we assume that the value function in the future time slot computed $\boldsymbol{b}$ is known thus there is no error at $\boldsymbol{b}$) and let $\boldsymbol{\alpha}'$ be the vector maximal at $\boldsymbol{b}'$. Therefore,

$$
\begin{aligned}
\epsilon \quad &\leq [\boldsymbol{\alpha}']'\boldsymbol{b}' - [\boldsymbol{\alpha}]\boldsymbol{b}' \\
&= [\boldsymbol{\alpha}']'\boldsymbol{b}' - [\boldsymbol{\alpha}]\boldsymbol{b}' + [\boldsymbol{\alpha}']'\boldsymbol{b} - [\boldsymbol{\alpha}']'\boldsymbol{b} \\
&\leq [\boldsymbol{\alpha}']'\boldsymbol{b}' - [\boldsymbol{\alpha}]\boldsymbol{b}' + [\boldsymbol{\alpha}]'\boldsymbol{b} - [\boldsymbol{\alpha}']'\boldsymbol{b} \\
&= [(\boldsymbol{\alpha}' - \boldsymbol{\alpha})]'(\boldsymbol{b} - \boldsymbol{b}') \\
&\leq ||[(\boldsymbol{\alpha}' - \boldsymbol{\alpha})]'||_\infty ||(\boldsymbol{b} - \boldsymbol{b}')||_1
\end{aligned}
$$

In the last equation above, we use Holder's inequality. Note that $||[(\boldsymbol{\alpha}' - \boldsymbol{\alpha})]'||_\infty$ represents the maximum difference in the costs that are achieved starting from a certain state and is given as $\zeta$. Note that $\zeta < \infty$ because the total number of time slots is finite and the costs in each decision epoch are bounded. Thus we can write the above inequality as

$$\epsilon \leq \zeta \delta_{\bar{B}[t]} \quad (28)$$

We now proceed to the overall error introduced by the Algorithm.

$$
\begin{aligned}
\epsilon_t \quad &= ||V^{\bar{B}[t]}(\ ,t) - V(\ ,t)||_\infty \\
&= ||\tilde{\Phi}_t V^{\bar{B}[t+1]}(\ ,t+1) - \Phi V(\ ,t)||_\infty \\
&= ||\tilde{\Phi}_t V^{\bar{B}[t+1]}(\ ,t+1) - \Phi_t V^{\bar{B}[t+1]}(\ ,t+1) + \Phi_t V^{\bar{B}[t+1]}(\ ,t+1) - \Phi V(\ ,t)||_\infty \\
&\leq ||\tilde{\Phi}_t V^{\bar{B}[t+1]}(\ ,t+1) - \Phi_t V^{\bar{B}[t+1]}(\ ,t+1)||_\infty + ||\Phi_t V^{\bar{B}[t+1]}(\ ,t+1) - \Phi V(\ ,t)||_\infty \\
&\leq \zeta \delta_{\bar{B}[t]} + \delta \epsilon_{t+1} \\
&= \zeta \delta_{\bar{B}[t]} + \delta \zeta \delta_{\bar{B}[t+1]} + \delta^2 \epsilon_{t+2} \\
&= \zeta \Omega(\bar{B}) \frac{1}{1-\delta}
\end{aligned}
$$

The above result can be extended to the finite horizon case. If $T$ is sufficiently large, then the value function achieved by the proposed policy and the exact policy will be close to $V^{\tilde{B}[t]}(,t)$ and $V(,t)$ respectively. If $\delta_{\bar{B}}$ is sufficiently small, then the proposed and the exact optimal policy will achieve very similar value function.

If the approximation error goes to zero, then the value function of the proposed and the exact optimal policy are the same. Based on the assumption that there is a unique optimal solution to (3), we can see that the proposed and the exact optimal policies will also be identical.

## 1.6  Appendix F

**Proof for Theorem 1.**

In Lemma 1 we showed that $V(\boldsymbol{b}, t)$ can be written as $\max_{\boldsymbol{\alpha}} \boldsymbol{\alpha}^* \boldsymbol{b}$. Since we are considering the space of models in this Theorem, we will define the value function on the space $\boldsymbol{L}$. Following Lemma 1 we can write

$$V((\boldsymbol{b}, \boldsymbol{m}), t) = \max_k \boldsymbol{\alpha}(k, \boldsymbol{m})^* \boldsymbol{b} \tag{29}$$

where $(\boldsymbol{b}, \boldsymbol{m}) \in \boldsymbol{L}$ and $\boldsymbol{\alpha}(k, \boldsymbol{m})$ is the $k^{th}$ alpha vector for model $\boldsymbol{m}$. We can assume that the same indexing is used for the alpha vectors across all the models. (Also, based on the definition of alpha vectors the total number of alpha vectors is the same across all the models.)

Consider a fixed belief $\boldsymbol{b}$ and a fixed model $\boldsymbol{m}$. $a^*((\boldsymbol{b}, \boldsymbol{m}), t)$ is the unique maximizer (except for a set of measure zero of models $\boldsymbol{m}$; this is based on the assumption). Let $k^*(\boldsymbol{m}, \boldsymbol{b})$ corresponds to the index of the corresponding optimal alpha vector. (Here we have assumed that the index $k^*(\boldsymbol{m}, \boldsymbol{b})$ is unique, but this assumption can be relaxed.) Based on the assumption that for a fixed $\boldsymbol{m}$ $\boldsymbol{b}$, $a^*((\boldsymbol{b}, \boldsymbol{m}), t)$ is a unique maximizer (except for a set of measure zero of models $\boldsymbol{m}$), we can conclude that the $\boldsymbol{\alpha}(k^*(\boldsymbol{m}, \boldsymbol{b}), \boldsymbol{m})^* \boldsymbol{b}$ is strictly better than other $[\boldsymbol{\alpha}(k, \boldsymbol{m})^* \boldsymbol{b}, ; \forall k \neq k^*(\boldsymbol{m}, \boldsymbol{b})$. Note that there may exist $\boldsymbol{m}, \boldsymbol{b}$ for which the maximizer $k^*(\boldsymbol{m}, \boldsymbol{b})$ is not unique. The measure of such a set is zero (as it will amount to finding $\boldsymbol{m}, \boldsymbol{b}$ such that $\boldsymbol{\alpha}(k^*(\boldsymbol{m}, \boldsymbol{b}), \boldsymbol{m})^* \boldsymbol{b} = \alpha(k, \boldsymbol{m})^* \boldsymbol{b}$ for some $k \neq k^*(\boldsymbol{m}, \boldsymbol{b})$), thus we can exclude these points.

If all the probability distributions defined in the model are continuous in $\boldsymbol{m}$, then $\alpha(k, \boldsymbol{m})$ is a continuous vector valued function of $\boldsymbol{m}$ for all $k$ as well. Therefore, the condition $\alpha(k, \boldsymbol{m})^* \boldsymbol{b}$ has to be strictly better than $\alpha(k, \boldsymbol{m})^* \boldsymbol{b}$, $\forall k \neq k^*(\boldsymbol{m}, \boldsymbol{b})$ in a neighborhood of $\boldsymbol{m}$. In fact, due to the continuity of $\alpha(k, \boldsymbol{m})^* \boldsymbol{b}$ in $\boldsymbol{b}$, this has to hold true for a neighborhood in the joint space $\boldsymbol{L}$. This implies that the optimal action $a^*$ stays fixed in the neighborhood as well. This proves the result.

Table 1: Comparison of the proposed policy with biennial policy.

| Risk Group | Metrics | DPSCREEN with self-detection | Proposed w/o self-detection | Biennial |
|---|---|---|---|---|
| Low | $E[N|R], E[\Delta|R], E[\Delta|R,D]$ | 0.21, 0.29, 12.36 | 0.42, 0.29, 12.36 | 0.5, 0.29, 12.36 |
| High | $E[N|R], E[\Delta|R], E[\Delta|R,D]$ | 0.22, 0.90, 12.13 | 0.38, 0.90, 12.13 | 0.5, 0.88, 11.8 |

## 1.7 Appendix G

**Further details on illustrative experiments:**

**Mammogram output:** The outcome of a mammogram is given in the form of a BIRADS (Breast Imaging Report and Data System) score $\{1, 2, 3, 4, 4A, 4B, 4C, 5, 6\}$, The outcome was considered positive if the BIRADS scores is 4 or above, in which case a biopsy was performed.

**Model Estimation:** We use independent normal priors for the parameters of the functions $p_{in}(x)$ and $p_{tr}(x)$. We compute the posterior (up to a constant) of the parameters in terms of the likelihood of the observed data (described above). We estimate the posterior distribution using the Metropolis Hastings method with a Gaussian random walk as the proposal distribution.

**Comparisons with Biennial Policies** In Table 3 we compare the performance of DPSCREEN (with and without self-examination) for Low and High risk groups against the current clinical policy of biennial screening. For both risk groups, the proposed policy achieves approximately the same expected delay as the benchmark policy while doing many fewer tests (in expectation). With self-examinations, the expected reduction in number of screens is 56-58% (depending on risk group); even without self-detection, the expected reduction in number of screens is 16-24% percent (depending on risk group).