[Reviews · NeurIPS 2017]

Reviewer 1



This paper presents DPSCREEN, a method for determining optimal screening policies for large classes of diseases. Compared to other work in the area, DPSCREEN takes into account both the patient's clinical (screening) history, in addition to patient features pertinent to the specific disease model. As a result, DPSCREEN is able to achieve personalization in the screening process, which yields efficiency gains over existing screening policies without negatively impacting the timeliness of disease detection. Overall I found this paper to be well-written and technically sound. The approach is well-motivated, considers pertinent related work, and provides a concise but thorough discussion of the mathematical basis of the algorithm, including computational complexity. DPScreen is validated on a large (45k patient) dataset from the Athena Health Network. This sufficiently demonstrates the improvements provided by personalization. The model is also validated across disease model types. The result is a paper that makes a clear technical contribution that also has the potential for practical impact as well.

Reviewer 2



The objective of the paper is to find the best policy for patient screening given the pertinent information. To provide the policy, a disease should be modeled as a finite state stochastic process. The policy is trying to minimize screening costs and delays. The authors propose an approximate solution that is computationally efficient. The experiments on simulated data related to breast cancer indicate that the proposed algorithm could reduce delays while keeping the same cost when compared to a trivial baseline Positives: + personalized screening is an important topic of research + the proposed algorithm is reasonable and well-developed + the results are promising + the paper is well written Negatives: - the proposed algorithm is purely an academic exercise. It assumes that the disease model of a given patient is known, which could be never assumed in practice. The main obstacle in personalized screening is not coming up with a policy when the model is known, but inferring the model from data. - the methodological novelty is not large - the experiments only compare the proposed algorithm to a baseline doing annual screening. Given the previous point about the disease model, an important question that should be studies experimentally is the robustness of the proposed method to inaccuracies in the disease model. In other words, how sensitive is the policy to uncertainties in the disease model. Overall, the topic is interesting and the proposed algorithm is reasonable. However, the methodological novelty is not large and the paper would need to do a better work to convince readers that this work is practically relevant.

Reviewer 3



The paper suggests a method for predicting appropriate screening time for breast cancer patients to minimize screening cost and delay cost. This is done in a personalized fashion by taking into account the personal medical history of the patient along with external information. The paper is generally well written and the proposed method shows significant improvement over existing methods.